# Propentofylline Improves Thiol-Based Antioxidant Defenses and Limits Lipid Peroxidation following Gliotoxic Injury in the Rat Brainstem

**DOI:** 10.3390/biomedicines11061652

**Published:** 2023-06-07

**Authors:** Deborah E. M. Baliellas, Marcelo P. Barros, Cristina V. Vardaris, Maísa Guariroba, Sandra C. Poppe, Maria F. Martins, Álvaro A. F. Pereira, Eduardo F. Bondan

**Affiliations:** 1Department of Veterinary Medicine, Cruzeiro do Sul University, São Paulo 08060070, Brazil; dbaliellas@gmail.com (D.E.M.B.); maisa_bizie@hotmail.com (M.G.); sandra.poppe@cruzeirodosul.edu.br (S.C.P.); fa3m@terra.com.br (M.F.M.);; 2Interdisciplinary Programs in Health Sciences, Institute of Physical Activity and Sport Sciences (ICAFE), Cruzeiro do Sul University, São Paulo 01506000, Brazil; crisvardaris@gmail.com; 3Graduate Program in Environmental and Experimental Pathology, University Paulista (UNIP), São Paulo 04057000, Brazil

**Keywords:** astrocyte, glial cells, xanthine, oxidative stress, free radicals, neurodegenerative, caffeine

## Abstract

Propentofylline (PROP) is a methylated xanthine compound that diminishes the activation of microglial cells and astrocytes, which are neuronal cells strongly associated with many neurodegenerative diseases. Based on previously observed remyelination and neuroprotective effects, PROP has also been proposed to increment antioxidant defenses and to prevent oxidative damage in neural tissues. Since most neurodegenerative processes have free radicals as molecular pathological agents, the aim of this study was to evaluate the antioxidant effects of 12.5 mg·kg^−1^·day^−1^ PROP in plasma and the brainstem of Wistar rats exposed to the gliotoxic agent 0.1% ethidium bromide (EB) for 7–31 days. The bulk of the data here demonstrates that, after 7 days of EB treatment, TBARS levels were 2-fold higher in the rat CNS than in control, reaching a maximum of 2.4-fold within 15 days. After 31 days of EB treatment, lipoperoxidation in CNS was still 65% higher than that in the control. Clearly, PROP treatment limited the progression of lipoperoxidation in EB-oxidized CNS: it was, for example, 76% lower than in the EB-treated group after 15 days. Most of these effects were associated with PROP-induced activity of glutathione reductase in the brainstem: the EB + PROP group showed 59% higher GR activity than that of the EB or control groups within 7 days. In summary, aligning with previous studies from our group and with literature about MTXs, we observed that propentofylline (PROP) improved the thiol-based antioxidant defenses in the rat brainstem by the induction of the enzymatic activity of glutathione reductase (GR), which diminished lipid oxidation progression and rebalanced the redox status in the CNS.

## 1. Introduction

Methylxanthines (MTXs) are purine-based alkaloids that are frequently found in highly consumed foods and beverages such as coffee, cacao, chocolate, and tea [1,2]. After uptake by the human body, MTXs are freely distributed in the bloodstream to different tissues, where they are absorbed at different rates. Among all absorbing tissues/organs, MTXs easily cross the blood–brain barrier to accumulate in several segments of the central nervous system (CNS) [3].

Many MTXs display substantial neuro-boosting properties, similar to that of caffeine, which were associated with important antioxidant and neuroprotective effects that significantly diminish oxidative stress in brain portions and neuronal circuits [4]. Moreover, MTXs were shown to modulate inflammatory immune responses, usually by controlling the release of pro-inflammatory cytokines and the attenuation of microglial activation [5]. On the molecular level, MTXs and their phosphate-derivatives are potential chelating agents of redox-active metals, such as ferrous and cupreous ions (Fe^2+^ and Cu^+^, respectively), which are catalysts for the formation of more aggressive reactive oxygen/nitrogen species (ROS/RNS) in biological systems [6]. Moreover, MTXs treatment was shown to affect major thiol-dependent antioxidant defenses, mainly glutathione (GSH), probably via redox-signaling cascades, such as the Keap1-Nrf2 and NF-kB pathways [7]. Thiol-dependent antioxidants, such as GSH, peroxiredoxin, and glutaredoxin, are direct scavengers of ROS/RNS, as well as substrates for major antioxidant enzymes, such as glutathione peroxidase (GPx), glutathione reductase (GR), and glutathione-S-transferase (GST) [8]. Figure 1 shows natural and synthetic MTXs.

The CNS, all brain regions included, is particularly sensitive to oxidative stress [8]. The animal brain is a highly (ATP) energy-demanding organ, normally supplied with an enormous amount of molecular oxygen (O_2_), engaged with intense mitochondrial activity, and it is particularly rich in polyunsaturated fatty acids, which, altogether, constantly expose this organ to harmful oxidative/nitrative conditions [9]. In an even worse scenario, the brain accumulates prooxidant iron ions for proper cognitive functions (although with distinguished distribution between brain regions) and surprisingly limited antioxidant capacities, especially in terms of the H_2_O_2_-removing enzyme catalase [10]. Therefore, ROS/RNS accumulation in brain regions (acute or chronic) is a cellular threat that, if not properly counteracted by local and systemic antioxidants, can cause significant neuronal damage [8]. From all subcellular sources of ROS/RNS, mitochondria are unquestionably the main organelles associated with oxidative/nitrative injuries to biomolecules during aging and neurodegenerative processes [11].

Ethidium bromide (EB) has been applied as a gliotoxin to impose oxidative conditions within brain regions, causing oligodendroglial and astrocytic death, severe demyelination (although “naked” axons remain preserved), blood–brain barrier impairments, and Schwann cell invasion, especially when injected in the white matter of the CNS [12]. Previous studies from our group have shown that propentofylline (PROP), a synthetic MTX, acts to (i) decrease astrocytic activation, thus reducing glial scar development following injury [12], (ii) increase both oligodendroglial and Schwann cell remyelination after 31 days, compared to untreated animals [13], and (iii) even reverse the neuronal dysfunction caused by demyelination induced by the diabetic state in Wistar rats [14]. Considering that most of the PROP-mediated healing processes involve redox (free radical) chemistry, we aimed here to investigate the effect of 12.5 mg·kg^−1^·day^−1^ PROP on biomarkers of oxidative stress in plasma and the brainstem regions (pons and mesencephalon) of Wistar rats treated with 0.1% EB (as a gliotoxin) after 7, 15, and 31 days. Based on previous behavioral and morphological data, we expected to observe significant antioxidant activity of PROP, both in plasma and the brain regions, in EB-injured animals.

## 2. Materials and Methods

### 2.1. Animals

Adult (4–5 months old) male Wistar rats were obtained from the Laboratory of Animal Resources, Paulista University (UNIP), São Paulo, Brazil, and were randomly distributed in a four-arm parallel group experiment [15]. Three sets of 24 animals each were initially used in experimental treatments, from which 21 samples were collected after 7 days of treatment; 23 samples were collected after the 15-day intervention; and 22 samples were collected after the 31-day intervention. The animals were kept under controlled light conditions (12 h light-dark cycle) and water and standard laboratory animal feed (52% carbohydrate, 21% protein, and 4% lipid; Nuvilab CR1, Nuvital, Curitiba, PR, Brazil) were provided ad libitum during the experimental period. All animal procedures were performed in accordance with the guidelines of the Committee on Care and Use of Laboratory Animal Resources and Brazilian Institutional Ethics Committee, Paulista University (protocol number 182/13, CEUA/ICS/UNIP).

This study presents four experimental groups: (i) control; (ii) treated with 0.1% ethidium bromide (EB); (iii) treated with 12.5 mg·kg^−1^·day^−1^ of propentofylline (PROP); and (iv) both EB and PROP treatments, in the same doses. All rats were anaesthetized with 2.5% thiopental (50 mg·kg^−1^) by an intraperitoneal route. After that, a burr-hole was made on the right side of the skull, 8 mm behind the frontoparietal suture for drug administration. Injections of 10 μL of 0.1% EB were performed into the cisterna pontis (an enlarged subarachnoid space below the ventral surface of the pons). The pons is a connective structure that links the base of the brain to animal spinal cord and is associated with unvoluntary tasks, such as the sleep–wake cycle and breathing [16]. Injections were performed freehand using a Hamilton syringe, fitted with a 35° angled polished 26-gauge needle into the cisterna pontis. Rats treated with propentofylline (PROP) received 12.5 mg·kg^−1^·day^−1^ of PROP (Agener União Química, São Paulo, SP, 20 mg·mL^−1^ solution) by an intraperitoneal route daily during the experimental period.

### 2.2. Plasma and Tissue Samples

The choice of the appropriate sampling method is known to be crucial for accurate haematological and clinico-biochemical measurements [17]. Blood samples were collected in EDTA-containing Vacutainer^®^ flasks for plasma isolation (after centrifugation for 5 min, at 4× *g*, RT). Heparin-coated tubes were avoided here, as the study aimed to measure indices involving iron metabolism or iron-chelating capacities. For biochemical analysis in the nervous tissue, all rats from experimental groups were euthanized and whole brains were collected at each of the sampling periods: 7, 15, and 31 days post-EB injection (p.i.). Pons and mesencephalon portions of the brainstem were cautiously removed, immediately frozen in liquid nitrogen, and stored in a −80 °C freezer for further analyses. Figure 2 sketches the experimental design of our study.

### 2.3. Biochemical Analyses

The brainstem (including pons and mesencephalon) of the animals was rapidly thawed and homogenized using a pestle (or a Potter apparatus) in 1–2 mL of 50 mM phosphate buffer, pH 7.4, then centrifuged for 10 min at 10,000× *g* and 4 °C. Tissue/cellular debris was discarded and homogenates were kept in an ice-water bath for immediate enzyme and other biochemical determinations.

Catalase activity was determined by H_2_O_2_ consumption for 5 min at 25 °C. In each assay, 10 µL of sample were added to the reaction system composed by 10 mM H_2_O_2_ in 50 mM KPO (potassium phosphate) buffer, pH 7.4, and absorbance was monitored at 240 nm (ε_240nm_ = 0.071 mM^−1^·cm^−1^). Catalase activity was expressed in mU_CAT_.mg protein^−1^ [18].

Glutathione reductase (GR) activities were measured based on the oxidation of β-NADPH (at 340 nm; ε = 6.2 × 10^3^ M^−1^·cm^−1^). In the presence of 0.25 mM NaN_3_ (for catalase inhibition), a 10 µL sample was mixed with 1 mM of oxidized glutathione (GSSG) and 0.12 mM β-NADPH in reaction buffer (143 mM sodium phosphate and 6.3 mM EDTA, pH 7.4), at 25 °C. GR activity was expressed in mU_GR_·mg protein^−1^ [19]. All protein analyses were performed using the Coomassie-blue method described by Bradford, 1976 [20]. 

The total antioxidant capacity of the nervous tissue was measured by the ferric-reducing activity method (FRAP) [21]. The FRAP method quantifies metal ligands in samples that form [Fe(L)]^n+^ complexes that restrain Fenton-type reactions and the formation of more aggressive radicals, such as the hydroxyl radical (HO^•^). We adapted the method by replacing the classic ferrous-chelating agent 2,4,6-tripyridyl-S-triazine (TPTZ) by its analog 2,3-bis(2-pyridyl)-pyrazine (DPP) [22]. Briefly, 10–20 µL of samples were mixed with 10 mM DPP (from a stock solution in 40 mM HCl) and 20 mM of FeCl_3_ in a 0.30 M acetate buffer (pH 3.6). Absorbance at 593 nm was recorded for 4 min to determine the rate of Fe^2+^-DPP complex formation and was compared to a standard curve. Total iron content in brain tissue was estimated by the modified colorimetric method based on the formation of a Fe^2+^: bipyridyl complexes [23]. 

Measurements of reduced (GSH) and oxidized glutathione contents (GSSG) applied the stoichiometric reaction of reduced thiol groups (-SH) with 5,5′-dithio-2-nitrobenzoic acid (DTNB) to form TNB, which was monitored spectrophotometrically at 412 nm in 5 mM phosphate buffer, with 5 mM EDTA, pH 7.5. For GSSG determination, the current GSH forms in samples were prevented from oxidation by adding 0.2 mM 2-vinylpyridine (2VP) and its excess was eliminated by the further addition of 2 mM triethylamine (TEA). Then, the reaction system was added to 4 mM DTNB, 1 mM β-NADPH, and 0.25 U.mL^−1^ of enzyme GR for (i) full reduction of GSSG forms to GSH (total glutathione content) and (ii) reaction of total GSH with DTNB for TNB formation and detection at 412 nm [24]. The GSH and GSSG contents in the brainstem samples were expressed in mmol.mg protein^−1^. Finally, the ratio between GSH and GSSG contents was calculated and presented here as the “reducing power” in samples (dimensionless), as shown in Equation (1).
Reducing power (RP) = [GSH]/[GSSG],(1)
where [GSH] is the reduced glutathione concentration in the samples and [GSSG] is the oxidized glutathione concentration in the samples.

Lipid oxidation was estimated by the method of thiobarbituric acid-reactive substances (TBARS). Briefly, 250 µL of samples reacted with 0.35% thiobarbituric acid, with 1% Triton X-100, in 0.25 mM HCl to produce a pinkish chromophore that was detected by absorbance at 535 nm. A standard curve with 1,1′,3,3′-tetraethoxypropane was used for TBARS determination. The formation of TBARS occurred in plastic microtubes in a boiling bath (100 °C), for 10 min [25]. All spectrophotometric determinations were performed in a microplate reader SpectraMax M2 (Silicon Valley, CA, USA).

### 2.4. Statistical Analyses

The software Jamovi version 1.1.5 was used to perform the normality tests—Shapiro–Wilk, skewness and kurtosis, generalized linear model (GLM), and post-hoc with correction of the error rate to the significance level by Fisher method.

Outliers were determined using the interquartile range method (IQR × 2.2) [26] and, when necessary, the data were winsorized [27]. The boxplot graphics were made using the following site: http://shiny.chemgrid.org/boxplotr/. The results were analyzed for the significance level and the effect size for each experiment performed. The significance level adopted was *p* ≤ 0.05 and the differences reported in the comparison of groups, times, and interactions were reported by the difference (diff.) between group means, considering the type of distribution presented. 

## 3. Results

For 31 days, lipoperoxidation was monitored in plasma and the brainstem tissue of experimental animals, with similar variation patterns in both matrices, depending on the treatment applied (Table 1). Higher levels of lipoperoxidation were especially found in the CNS following EB treatment. After 7 days, the TBARS levels were 2-fold higher in the rat CNS than in saline administered rats (control), reaching a maximum of 2.4-fold within 15 days. After 31 days of EB treatment, lipoperoxidation was still 65% higher than that of the control in the brainstem, although these levels normalized in plasma. In agreement, levels of lipoperoxidation were also maximized in plasma after 15 days of EB treatment. Compared to control, the TBARS levels in plasma of EB-treated animals were 54% and 60% higher in days 7 and 15, respectively (Table 1). No variation in lipoperoxidation levels was observed with propentofylline treatment (PROP) in rats, either in plasma or CNS. Interestingly, by combining EB and PROP treatments, lipoperoxidation also showed similar patterns in plasma and brain tissues with no significant variation within 7 days and a minor increase in day 15 (50.4% and 37.5%, respectively), reaching baseline levels after 31 days (Table 1). Comparing the EB + PROP and EB groups, the lipoperoxidation levels in the EB groups were significantly higher after 15 days (76%).

On the other hand, both EB and PROP (or their combination) caused a major disruption of iron homeostasis in the CNS (Table 1). Maximum levels of “free” iron were measured in the brainstem after 15 days (almost 5-fold higher, independent of the treatment). Between the groups, no differences were observed along the 31 days of evaluation, except for PROP-treated animals after 7 days. The iron content in the brainstem of PROP group was identical to that found in controls after 7 days. Imbalances in iron homeostasis were only observed afterwards, as in all other treated groups (Table 1).

Figure 3 illustrates the antioxidant activities in the brainstem after 7 days of EB, PROP and EB + PROP combined treatments. Although catalase activities were unaltered in all groups (compared to control, Figure 3B), significant changes were observed in FRAP and GR activities (Figure 3A,C, respectively). EB treatment drastically dropped FRAP activity in the nervous tissue after 7 days (−75%, Figure 3A). Interestingly, PROP administration showed a tendency of diminished FRAP activities in the brainstem, but statistically this was not not different (*p* = 0.058; Figure 3A). Undoubtedly, combined EB + PROP treatment reverted the lower FRAP activities found in single EB-treated samples back to control levels (Figure 3A). Regarding GR activities, EB treatment did not affect the enzyme activity in the CNS, although 59% higher GR activity was measured in the EB + PROP group, compared to control (Figure 3C). Single PROP treatment only showed tendencies of higher GR activity compared to control (*p* = 0.054, Figure 3C).

Concerning thiol-based antioxidant defenses in the CNS, Figure 4 presents levels of reduced (GSH) and oxidized glutathione (GSSG), as well as their ratio expressed as the “reducing power” between experimental groups after 7 days (Figure 4A–C). Minor changes were observed in GSH contents in the brainstem upon the experimental conditions here, except for differences between EB (−65%) and PROP treatments (Figure 4A). Notably, PROP treatment induced significantly higher levels of GSSG, compared to all other experimental groups or control (Figure 4B). GSSG contents in PROP group were 163%, 77%, and 152% higher than control, EB, and EB + PROP groups, respectively (Figure 4B). Finally, the reducing power was significantly diminished with both EB (−60%) and PROP treatments (−65%) but reestablished upon treatment with both EB and PROP combined (Figure 4C).

## 4. Discussion

Apart from their renowned stimulation effects on the CNS, MTXs have been long suggested as inducers of antioxidant responses in many biological systems [28]. Here, we observed that the synthetic MTX, propentofylline (PROP), also increased antioxidant protection in the brainstem of rats exposed to the gliotoxic agent EB, especially in terms of thiol-based defenses: levels of reduced GSH, redox balance (GSH/GSSG ratio), and GR activities. The data here were in full agreement with other studies from our group that showed that PROP improved oligodendroglial and Schwann cell remyelination and even diminished neuronal dysfunction in chemically induced diabetic Wistar rats [13,14]. These neurodegenerative processes are notably mediated by pathological overproduction of reactive oxygen and nitrogen species (ROS/RNS) [29,30]. 

Regarding the time course of brainstem injury, previous works showed that EB-treated rats presented demyelinating lesions in the pons and mesencephalon after 7–11 days, which suggested that these regions were extensively exposed to oxidative conditions [31]. Following the progression of the healing process, thinly remyelinated axons could be significantly seen at day 15 [13]. Nonetheless, morphological differences between rats treated or not treated with PROP were clearer from day 21 p.i., as rats treated with PROP presented a greater proportion of oligodendrocyte remyelinated axons compared to the untreated ones. The time course of these previously observed remyelination processes perfectly matched with the progression of lipid oxidation in brainstem and plasma, as shown in Table 1.

Although the mechanism has not been fully unveiled, Table 1 shows that EB-induced injury in rat brainstem was associated with higher release of “free” (labile) iron ions, suggesting a disruption of iron homeostasis [32]. Labile iron ions, when not properly restrained by stocking proteins, such as ferritin and transferrin, catalyze the formation of harmful ROS/RNS, such as hydroxyl (HO^•^) and alkoxyl radicals (LO^•^) [33]. This is the main mechanism of the notorious participation of iron ions as biological prooxidant agents [34]. After 7 days of EB administration in the brainstem, the iron concentration was severely increased, a harmful effect that peaked after 15 days, and persisted until 31 days of treatment. The concomitant administration of PROP could not limit iron release in the CNS. In fact, PROP also promoted iron release in a way that was similar to EB-injection within the brainstem. Nevertheless, these results suggest that the chelation of labile iron ions—a preventive antioxidant event—was not the proper mechanism by which PROP protected rat CNS exposed to EB. 

Lipid oxidation was inhibited by PROP in EB-treated rats, as shown in Table 1. Interestingly, an identical pattern of lipid oxidation was reproduced in the plasma of animals, suggesting that brain injury generated prooxidant agents (probably the labile iron ions themselves) that disseminated oxidative stress conditions in the plasma and all over the body of the animals. Unfortunately, we did not monitor other biomarkers of oxidative stress in plasma to confirm that.

Data shown in Figure 3 and Figure 4 demonstrate that combined EB + PROP treatments presented similar indices of antioxidant capacity as control groups, especially in terms of FRAP scores and the reducing power (Figure 3A and Figure 4C, respectively). Based on PROP effects over thiol-based antioxidant defenses (Figure 3C and Figure 4A–C), it is very plausible that this positive effect was triggered by the induction of the enzyme glutathione reductase (GR; Figure 3C), which recycles oxidized GSSG molecules back to their reduced form, GSH. Reduced GSH is extensively used for free radical removal (as scavengers of ROS/RNS) and as a conjugation substrate for xenobiotic elimination from animals’ bodies (via glutathione-S-transferase; GST) [35]. Interestingly, the treatment with pentoxifylline (an analog of PROP) in mice injected with B16F10 melanoma cells (high energy demanding cells) significantly reduced oxidative stress by also attenuating the altered levels of GSH and lipid peroxides [36]. Accordingly, the natural MXT caffeine also affected GSH levels by increasing glutathione S-transferase activity (GST), causing, in this case, GSH depletion and higher lipid peroxidation indices in high-energy-demanding B16F1 melanoma cells [36]. Moreover, theobromine, another natural MTX (Figure 1), also promoted neuroprotection to the rat brain in a transient global cerebral ischemia-reperfusion model, which was notoriously mediated by ROS/RNS [37]. Despite the bulk of data about the close relationship between MTX activation of the GSH-based antioxidant defenses in brain regions, the proper mechanism is still under debate [38]. Nevertheless, but not surprisingly, the most consistent data about MTX effect on GSH biosynthesis involve the activation of the transcription factors Nrf2, NF-κB, and AP-1, which are the most redox-responsive signaling cascades in animals and humans [39,40,41]. Figure 5 depicts the oxidative stress conditions observed in rat brainstem (with details of prooxidant and antioxidant markers), treated with EB in the presence or absence of PROP.

## 5. Conclusions

In summary, aligning with previous studies from our group and with literature about MTXs, we observed that propentofylline (PROP) improved the thiol-based antioxidant defenses in the rat brainstem by the induction of the enzymatic activity of glutathione reductase (GR), which diminished lipid oxidation progression and rebalanced the redox status in the CNS.

## Figures and Tables

**Figure 1 biomedicines-11-01652-f001:**
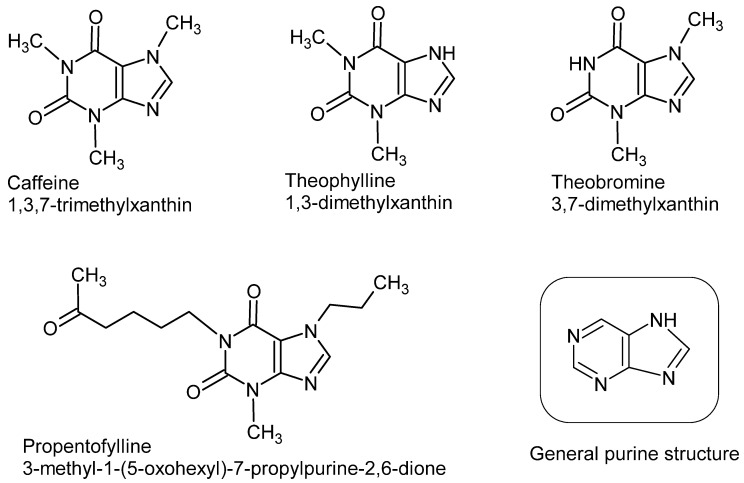
Chemical structures of common purines, caffeine, theophylline, theobromine, and the synthetic methylxanthine, propentofylline.

**Figure 2 biomedicines-11-01652-f002:**
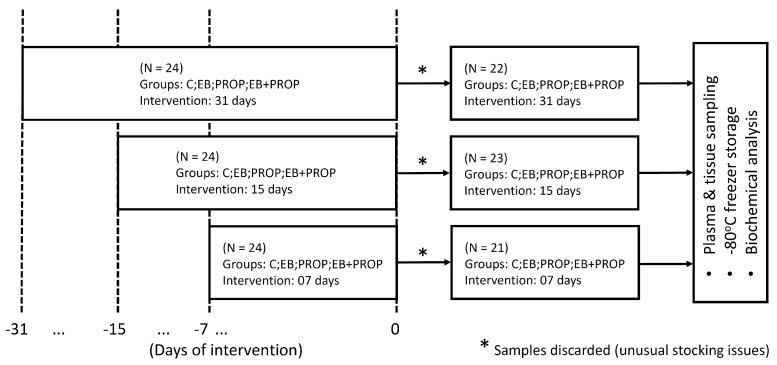
Experimental design of the study.

**Figure 3 biomedicines-11-01652-f003:**
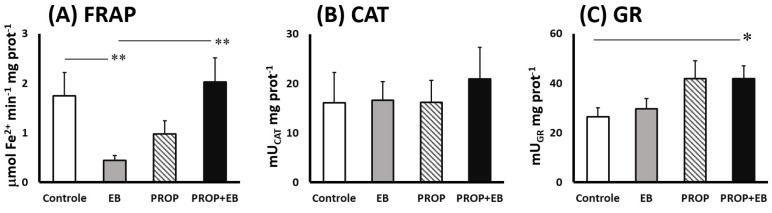
Antioxidant activities in brainstem of Wistar rats after 7 days of treatment (i.p.) with saline solution (control), 0.1% ethydium bromide (EB), 12.5 mg propentofylline·kg^−1^·day^−1^ (PRO), or both (EB + PROP): (**A**) ferric-reducing activity (FRAP); (**B**) catalase activity (CAT); and (**C**) glutathione reductase activity (GR). (* *p* < 0.05; ** *p* < 0.01).

**Figure 4 biomedicines-11-01652-f004:**
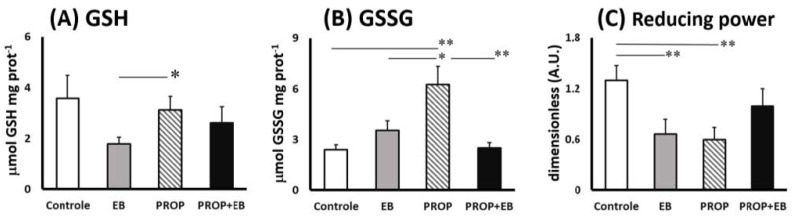
Thiol-based antioxidant defenses in brainstem of Wistar rats after 7 days of treatment (i.p.) with saline solution (control), 0.1% ethydium bromide (EB), 12.5 mg propentofylline·kg^−1^·day^−1^ (PRO), or both (EB + PROP): (**A**) reduced glutathione content (GSH); (**B**) oxidized glutathione content (GSSG); and (**C**) GSH/GSSG ratio, renowned as “reducing power”. (* *p* < 0.05; ** *p*< 0.01).

**Figure 5 biomedicines-11-01652-f005:**
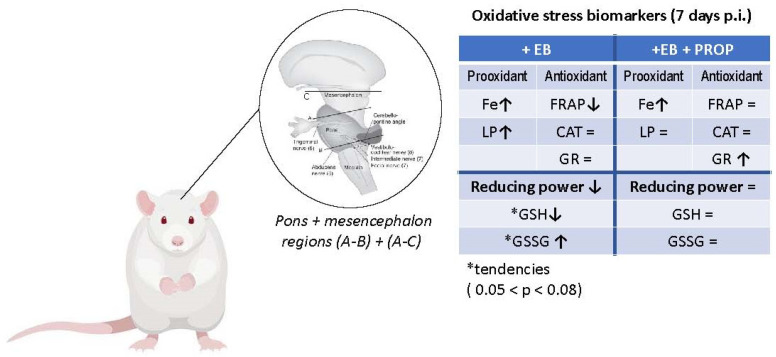
Graphic summary of pro- and antioxidant changes in brainstem (Pons + mesencephalon) of Wistar rats after 7 days of i.p. treatment with 0.1% ethydium bromide (EB) in the presence or absence of 12.5 mg propentofylline·kg^−1^·day^−1^ (PROP). Where Fe = iron content; LP = lipid oxidation; FRAP = ferric-reducing activity; CAT = catalase activity; GR = glutathione reductase activity; GSH = reduced glutathione; GSSG = oxidized glutathione; reducing power= GSH/GSSG ratio; ↑ = statistical increase; ↓ = statistical decrease; and * (statistical tendency 0.05 < *p* < 0.08). Adapted from P. Brodal, 2014 [42].

**Table 1 biomedicines-11-01652-t001:** Levels of lipid oxidation and total iron content in plasma or brainstem tissue of rats treated with ethidium bromide (EB) and/or propentofylline (PROP) for 7, 15, and 31 days. (* *p* < 0.05; ** *p*< 0.01).

	Control	EB	PROP	PROP + EB
(i) Plasma				
TBARS (μmol·mL^−1^)				
7 d	13.1 ± 2.4	20.2 ± 2.4 (*)	11.8 ± 0.9	15.4 ± 1.5
15 d	n.d.	21.0 ± 2.9 (*)	15.1 ± 2.6	19.7 ± 6.4 (*)
31 d	n.d.	10.8 ± 1.1	12.9 ± 2.6	12.4 ± 2.1
(ii) Brainstem tissue				
TBARS (μmol·mg prot^−1^)				
7 d	0.64 ± 0.04	1.28 ± 0.27 (**)	0.54 ± 0.06	0.72 ± 0.09
15 d	n.d.	1.55 ± 0.28 (**)	0.63 ± 0.05	0.88 ± 0.05 (*)
31 d	n.d.	1.06 ± 0.13 (**)	0.52 ± 0.05	0.72 ± 0.07
Iron content (μg·mg prot^−1^)				
7 d	88.2 ± 10.5	279.0 ± 60.6 (**)	75.8 ± 24.7	214.5 ± 52.4 (**)
15 d	n.d.	413.4 ± 31.6 (**)	433.3 ± 69.2 (**)	409.7 ± 41.8 (**)
31 d	n.d.	285.3 ± 50.8 (**)	261.0 ± 35.2 (**)	279.6± 30.8 (**)

n.d. = not determined.

## Data Availability

The data presented in this study are available on request from the corresponding authors. The data are not publicly available due to potential patent application.

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
