# Peer review of "Propentofylline Improves Thiol-Based Antioxidant Defenses and Limits Lipid Peroxidation following Gliotoxic Injury in the Rat Brainstem"

_biomedicines, 2023, doi:10.3390/biomedicines11061652_

Round 1

Reviewer 1 Report

we read with interest the article by Baliellas discussing the antioxidant activity of Propentofylline post-ethidium bromide injection.

the study utilized a battery of biochemical assessments of lipid peroxidation (A), ferric-reducing activity (FRAP); (B) catalase activity (CAT); and (C) glutathione reductase activity after 7, 15, and 31 days. where they showed that these activities were enhanced drawing a conclusion that PROP) improved the thiol-based antioxidant defenses in the rat brainstem

The study has several weaknesses:

the authors proposed 21 animals distributed among 4 groups, the N is so low to justify the results.

the N of animals is not enough for the 3-time points to be assessed: are they doing n=2/ time point to be sacrificed? the whole study number doesn't make sense.

the biochemical tests utilized are not specific but rather they measure whole oxidation rather than specific to what the author is mentioning. they need to do lipid peroxidation Western blotting as well as doing catalase levels, GSH levels using Immunologica;l assessment.

the work did not assess levels of astrocytic activation changes post-EB injection, GFAP, Iba-1, and NeuN should be evaluated via IF analysis.

these concerns represent major Serious flaws.

Author Response

First of all, we would like to thank Reviewer #1 for his/her comments and criticisms that undoubtedly improved the quality and literacy of our study. Accordingly we corrected our MS and this updated version brings these alterations marked in BLUE.

(x) English language fine. No issues detected.

Answer: Thank you. However, due to some questions from other referees, we decided to double-check the English content of this updated version of our MS.

We read with interest the article by Baliellas discussing the antioxidant activity of Propentofylline post-ethidium bromide injection: the study utilized a battery of biochemical assessments of lipid peroxidation (A), ferric-reducing activity (FRAP); (B) catalase activity (CAT); and (C) glutathione reductase activity after 7, 15, and 31 days. where they showed that these activities were enhanced drawing a conclusion that PROP) improved the thiol-based antioxidant defenses in the rat brainstem

The study has several weaknesses:

(1) the authors proposed 21 animals distributed among 4 groups, the N is so low to justify the results: the N of animals is not enough for the 3-time points to be assessed: are they doing n=2/ time point to be sacrificed? the whole study number doesn't make sense.

Answer: We fully agree with reviewer #1. As it stands, the methodology wrongly describes the total sample size used in the experiments here. In fact, the whole experiment involved 03 sets of 24 animals each: one set running for 7 days, another one for 15 days, and the third set maintained for 31 days. The original MS was planned to present only the results after 7 days of EB administration, but we decided afterwards to include some relevant biochemical evidence collected in further periods, in order to reinforce the harmful oxidative conditions imposed by EB-treatment. Unfortunately, we forgot to correct the updated sample size. Thanks for detecting this mistake.

(2) the biochemical tests utilized are not specific but rather they measure whole oxidation rather than specific to what the author is mentioning. they need to do lipid peroxidation Western blotting as well as doing catalase levels, GSH levels using Immunologica;l assessment.

Answer: As a consensus in the field, whenever approaching oxidative stress in biological samples, a complete study should approach three lines of evidence: (i) prooxidant events that  increased ROS/RNS production (or the detection of the reactive species themselves, as correctly suggested by reviewer #1); (ii) antioxidant responses, reflecting how the biological system copes with the oxidative conditions imposed; and (iii) levels of oxidation/nitration in biomolecules, indicating the efficiency of the antioxidant system to inhibit oxidation events in tissues and/or cells. Accordingly, we properly targeted here: (i) prooxidant events, by measuring Fe2+/3+ ions content and the reducing power in plasma and brainstem tissues (revealing the oxidative microenvironment within); (ii) antioxidant responses of FRAP, catalase (CAT), and glutathione redutase (GR) activities, as well as reduced glutathione content in brainstem (GSH); and (iii) oxidation products, by measuring TBARS and oxidized glutathione (GSSG) contents. However, we have to agree with reviewer #1 that those methods are NOT the most accurate ones available in the literature, e.g. TBARS assay to detect lipid peroxidation in biological samples [Hu C, Wang M, Han X. Shotgun lipidomics in substantiating lipid peroxidation in redox biology: Methods and applications. Redox Biol. 2017 Aug;12:946-955. doi: 10.1016/j.redox.2017.04.030. Epub 2017 Apr 24. PMID: 28494428; PMCID: PMC5423350]. On the other hand, we had to consider the afordable methods available in our laboratory that are still robust and well-accepted in the scientific literature. In time, the applied method for GSH/GSSG detection, based on DTNB chemistry, is very sensitive and reproductive, which has already been published in Nature Protocols, a renowned journal for detailed methods [Rahman, I., Kode, A., & Biswas, S. K. Assay for quantitative determination of glutathione and glutathione disulfide levels using enzymatic recycling method. Nature Protocols 2006 1(6): 3159-3165. doi: 10.1038/nprot.2006.378].

(3) the work did not assess levels of astrocytic activation changes post-EB injection, GFAP, Iba-1, and NeuN should be evaluated via IF analysis.

Answer: In fact, we decided to present the histological and structural data in a different paper, in order to extend the structural description and explore the morphological aspects (with details) for a more specific public. Herewith, we decided to discuss more deeply the redox biochemistry of EB-treated brainstem tissues and the protective effects of propentofylline, including the possible mechanisms involved. We have already published papers describing major morphological or biochemical observations upon PROP effetcs:

Bondan EF, Martins MF, Dossa PD, Viebig LB, Cardoso CV, Martins JL Júnior, Bernardi MM. Propentofylline reduces glial scar development following gliotoxic damage in the rat brainstem. Arq Neuropsiquiatr. 2016 74(9):730-736. doi: 10.1590/0004-282X20160108. PMID: 27706422.

Bondan EF, Martins Mde F, Menezes Baliellas DE, Monteiro Gimenez CF, Castro Poppe S, Martha Bernardi M. Effects of propentofylline on CNS remyelination in the rat brainstem. Microsc Res Tech. 2014 77(1):23-30. doi: 10.1002/jemt.22308. Epub 2013 Nov 1. PMID: 24185688.

Bondan EF, Martins Mde F, Bernardi MM. Propentofylline reverses delayed remyelination in streptozotocin-induced di-abetic rats. Arch Endocrinol Metab. 2015 59(1):47-53. doi: 10.1590/2359-3997000000009. PMID: 25926114.  

Reviewer 2 Report

1) The abstract should be significantly improved. The abstract should reflect the methods and specific results obtained by the authors. 2) Introduction is uninformative. The introduction does not introduce the reader to the essence of the problem. 3) For materials and methods, an experimental scheme should be drawn that will reflect the experimental groups of animals and the manipulations that were performed with them. 4) Conversely, the discussion is overly speculative and needs to be shortened.

The English language should be greatly improved.

Author Response

First of all, we would like to thank Reviewer #2 for his/her comments and criticisms that undoubtedly improved the quality and literacy of our study. Accordingly we corrected our MS and this updated version brings these alterations marked in BLUE.

(x) English very difficult to understand/incomprehensible

Answer: Despite the positive outcomes from the other reviewers, we decided to check English fluency once again. We hope that this 2nd version is more fluent and coherent in terms of English Language. Thank you.  

(1) The abstract should be significantly improved. The abstract should reflect the methods and specific results obtained by the authors.

ANSWER:. As suggested by Reviewer #2, we have fully reformulated the Abstract of our MS.

2) Introduction is uninformative. The introduction does not introduce the reader to the essence of the problem.

ANSWER:. As suggested by Reviewer #2, we have fully reformulated the Introduction session of our MS, including new or repositioned references (marked in blue). We accordingly included one additional paragraph (lines 44-53) to explain how MTXs potentially affect antioxidant defenses, specially thiol-dependent defenses, in different biological systems.

3) For materials and methods, an experimental scheme should be drawn that will reflect the experimental groups of animals and the manipulations that were performed with them.

ANSWER:. In agreement with Reviewer #2 suggestions, Figure 2 now presents the exerimental design of our study. 

4) Conversely, the discussion is overly speculative and needs to be shortened.

ANSWER:. As suggested by Reviewer #2, we tried to focus the discussion on the results we properly obtained, in a more concise and straight-to-point discussion. Nevertheless, we still think it is important to address the possible mechanisms involved, despite we did not fully evidence them herewith.

Reviewer 3 Report

The manuscript presented from Deborah E.M. Baliellas et al., entitled "Propentofylline improves thiol-based antioxidant defenses and limits lipid peroxidation following gliotoxic injury in the rat brainstem" is original and interesting, well written. However there are some critical point that should be improved: - The authors should extend the introduction, in particular explaning the role of glutathione and antioxidant in the context of oxidative stress, also mentioned other model organisms. - The authors showed that GSH and GSSG can be altered, but is not clear if they detect an increase of ROS. I would suggest the authors to use ROS probe (CELL-Rox) to detect the level of ROS in brain cells. For protocol they can use (Cacialli et al., Nature Communication 2021). - For results they should furnish negative control and the number of animal used.                

Author Response

First of all, we would like to thank Reviewer #3 for his/her comments and criticisms that undoubtedly improved the quality and literacy of our study. Accordingly we corrected our MS and this updated version brings these alterations marked in BLUE.

(x) English language fine. No issues detected

Answer: Thank you. Nevertheless, we decided to double-check the English content of this updated version of our MS, based on comments from other reviewers.

(1) The manuscript presented from Deborah E.M. Baliellas et al., entitled "Propentofylline improves thiol-based antioxidant defenses and limits lipid peroxidation following gliotoxic injury in the rat brainstem" is original and interesting, well written.

ANSWER:. We are grateful to Reviewer #3 for his/her positive appreciation of our work.

(2) However there are some critical point that should be improved: - The authors should extend the introduction, in particular explaning the role of glutathione and antioxidant in the context of oxidative stress, also mentioned other model organisms.

ANSWER:. In agreement with suggestions from Reviewer #2, we accordingly included one additional paragraph (lines 44-53) to explain how MTXs potentially affect antioxidant defenses, specially thiol-dependent defenses, in different biological systems.

(3) The authors showed that GSH and GSSG can be altered, but is not clear if they detect an increase of ROS. I would suggest the authors to use ROS probe (CELL-Rox) to detect the level of ROS in brain cells. For protocol they can use (Cacialli et al., Nature Communication 2021).

Answer: As a consensus in the field, whenever approaching oxidative stress in biological samples, a complete study should approach three lines of evidence: (i) prooxidant events that  increased ROS/RNS production (or the detection of the reactive species themselves, as correctly suggested by reviewer #1); (ii) antioxidant responses, reflecting how the biological system copes with the oxidative conditions imposed; and (iii) levels of oxidation/nitration in biomolecules, indicating the efficiency of the antioxidant system to inhibit oxidation events in tissues and/or cells. Accordingly, we properly targeted here: (i) prooxidant events, by measuring Fe2+/3+ ions content and the reducing power in plasma and brainstem tissues (revealing the oxidative microenvironment within); (ii) antioxidant responses of FRAP, catalase (CAT), and glutathione redutase (GR) activities, as well as reduced glutathione content (GSH); and (iii) oxidation products, by measuring TBARS and oxidized glutathione (GSSG) contents. However, we have to agree with reviewer #1 that those methods are NOT the most accurate ones available in the literature, e.g. TBARS assay to detect lipid peroxidation in biological samples [Hu C, Wang M, Han X. Shotgun lipidomics in substantiating lipid peroxidation in redox biology: Methods and applications. Redox Biol. 2017 Aug;12:946-955. doi: 10.1016/j.redox.2017.04.030. Epub 2017 Apr 24. PMID: 28494428; PMCID: PMC5423350]. On the other hand, we had to consider the afordable methods available in our laboratory that are still robust and well-accepted in the scientific literature. In time, the applied method for GSH/GSSG detection, based on DTNB chemistry, is very sensitive and reproductive, which has already been published in Nature Protocols, a renowned journal for detailed methods [Rahman, I., Kode, A., & Biswas, S. K. Assay for quantitative determination of glutathione and glutathione disulfide levels using enzymatic recycling method. Nature Protocols 2006 1(6): 3159-3165. doi: 10.1038/nprot.2006.378].

(4) For results they should furnish negative control and the number of animal used.               

Answer: We fully agree with reviewer #3. As it stands, the methodology wrongly describes the total sample size used in the experiments presented here. In fact, the whole experiment involved 03 sets of 24 animals each: one set running for 7 days, another one for 15 days, and the third set maintained for 31 days. The original MS here was planned to present only the results of the 7-day EB intervention, but we decided afterwards to include biochemical data from further periods, in order to reinforce the harmful oxidative conditions imposed by EB-treatment. Unfortunately, we forgot to correct the updated sample size. Thanks for detecting this mistake. We understand the importance of negative controls when presenting biochemical-physiological effects in living organisms, but we opted here to, at least, demonstrate the single-agent effects (harmful effects from EB, or protective action by PROP) versus their combined effects (induced harmful circumvented by further protective action).

Round 2

Reviewer 1 Report

accept

Author Response

Thanks for your criticisms and suggestions that unquestionably improved the quality of our paper.

Reviewer 2 Report

In principle, my comments are taken into account. I did not find a link to figure 2 in the article.

Author Response

Thanks for your comments and suggestions that unquestionably improved the quality of our paper. 

We reinforced the link to Figure 2 in this updated version of our MS (line 129).

Reviewer 3 Report

The authors improved the quality of the manuscript.

Author Response

(The authors gave the same response as above.)
